# Community of Bark and Ambrosia Beetles (Coleoptera: Curculionidae: Scolytinae and Platypodinae) in Agricultural and Forest Ecosystems with Laurel Wilt

**DOI:** 10.3390/insects13110971

**Published:** 2022-10-22

**Authors:** Kevin R. Cloonan, Wayne S. Montgomery, Teresa I. Narvaez, Daniel Carrillo, Paul E. Kendra

**Affiliations:** 1Subtropical Horticulture Research Station, USDA-ARS, 13601 Old Cutler Road, Miami, FL 33158, USA; 2Tropical Research and Education Center, University of Florida, 18905 SW 280 ST, Homestead, FL 33031, USA

**Keywords:** chemical ecology, essential oil lures, ethanol lures, *Euwallacea perbrevis*, invasive species, kairomones, *Persea americana*, *Persea palustris*, pest monitoring, *Xyleborus glabratus*

## Abstract

**Simple Summary:**

Redbay ambrosia beetle (RAB), a wood-borer native to Southeast Asia, was first detected in North America in 2002 in Georgia, USA. The beetle carries a fungal symbiont that causes laurel wilt, a lethal disease of trees in the family Lauraceae. RAB is now established in 12 southeastern states where laurel wilt has caused widespread mortality of native forest trees, including redbay, swampbay, and silkbay. In Florida, laurel wilt also impacts avocado, but in contrast to the situation in forests, RAB is detected at very low levels in affected groves. Other species of ambrosia beetle have now acquired the fungal pathogen and contribute to the spread of laurel wilt. To better understand the beetle communities in different ecosystems exhibiting laurel wilt, parallel trapping tests were conducted in an avocado grove and a swampbay forest in Florida. Traps were baited with ethanol lures (the best general attractant for ambrosia beetles), essential oil lures (the best attractants for RAB), and combinations of these lures, resulting in captures of 20 species. This study (1) documents differences in beetle diversity and population levels at the two sites, and (2) identifies the best lures or lure combinations for detection of different beetle species.

**Abstract:**

Redbay ambrosia beetle, *Xyleborus glabratus*, is an invasive wood-boring pest first detected in the USA in 2002 in Georgia. The beetle’s dominant fungal symbiont, *Harringtonia*
*lauricola*, causes laurel wilt, a lethal disease of trees in the Lauraceae. Over the past 20 years, *X. glabratus* and laurel wilt have spread to twelve southeastern states, resulting in high mortality of native *Persea* species, including redbay (*P. borbonia*), swampbay (*P. palustris*), and silkbay (*P. humilis*). Laurel wilt also threatens avocado (*P. americana*) in south Florida, but in contrast to the situation in forests, *X. glabratus* is detected at very low levels in affected groves. Moreover, other species of ambrosia beetle have acquired *H. lauricola* and now function as secondary vectors. To better understand the beetle communities in different ecosystems exhibiting laurel wilt, parallel field tests were conducted in an avocado grove in Miami-Dade County and a swampbay forest in Highlands County, FL. Sampling utilized ethanol lures (the best general attractant for ambrosia beetles) and essential oil lures (the best attractants for *X. glabratus*), alone and in combination, resulting in detection of 20 species. This study documents host-related differences in beetle diversity and population levels, and species-specific differences in chemical ecology, as reflected in efficacy of lures and lure combinations.

## 1. Introduction

Laurel wilt is a systemic vascular disease of trees and shrubs in the family Lauraceae caused by the fungus *Harringtonia lauricola* T.C. Harr., Fraedrich & Aghayeva (Ophiostamatales: Ophiostomataceae) [1,2] (previously *Raffaelea lauricola* [3]). *Harringtonia lauricola* is a nutritional fungal symbiont associated with the wood boring redbay ambrosia beetle, *Xyleborus glabratus* Eichhoff (Coleoptera: Curculionidae: Scolytinae) [4]. Conidia of the fungus are housed inside specialized cuticular pouches called mycangia located at the base of the mandibles [5] and female beetles inoculate their brood galleries with these conidia during colonization of host trees [6]. Beetle larvae and adults feed directly on *H. lauricola* not the host wood itself [6], and the developing *H. lauricola* fungus does not directly kill the host tree.

Mortality from laurel wilt disease results from the host defensive responses that attempt to restrict the movement of *H. lauricola* [7], with parenchymal tyloses ultimately blocking the xylem vessels, impeding water transport, and leading to tree death [8]. In the USA laurel wilt is particularly lethal to trees in the genus *Persea* and documented suscepts include redbay, [*Persea borbonia* (L.) Spreng.] [5,9], swamp bay [*P. palustris* (Raf.) Sarg.] [10,11], silkbay (*P. humilis* Nash) [12], and avocado (*P. americana* Mill.) [13]. The disease is less severe, though may still cause wilting and death, in other genera, including sassafras [*Sassafras albidum* (Nutt.) Nees] [10,11], California bay laurel [*Umbellularia californica* (Hook. & Arn.) Nutt.] [1], northern spicebush [*Lindera benzoin* (L.)] [5,14], pondberry [*L. melissifolia* (Walter) Blume] [15], pondspice [*Litsea aestivalis* (L.) Fern.] [15,16], bay laurel (*Laurus nobilis* L.) [17], Gulf licaria, [*Licaria triandra* (Sw.) Kosterm.] [18], camphor tree [*Cinnamomum camphora* (L.) J. Presl.] [19,20], and lancewood [*Nectandra coriacea* (Sw.) Griseb.] [10,11].

The redbay ambrosia beetle is native to Southeast Asia [21] and was first detected in North America in Port Wentworth, Georgia, USA in 2002 [22,23] as a single introduction event [4]. By 2003 laurel wilt was detected in redbay forests of Georgia and South Carolina [6], and since then laurel wilt has spread rapidly throughout the Atlantic and Gulf coastal plains [1,20,24,25,26,27,28,29]. As of April 2022 laurel wilt has been detected in 12 states and has killed an estimated 300,000 redbay trees [4], with greater than 90 percent tree-loss reported in some forested areas of the USA [5,30,31,32,33]. Additionally, laurel wilt is currently a threat to the $13.7 million dollar avocado industry in Florida [34] and has killed over 200,000 commercial trees since its introduction into Miami-Dade County in 2010 [13,35,36,37,38,39]. Although laurel wilt is prevalent in the commercial avocado orchards of south Florida, *X. glabratus* adults are rarely captured or detected in this agrosystem [40,41,42,43].

The absence of significant *X. glabratus* populations in avocado suggests that alternative ambrosia beetle species have acquired *H. lauricola* through the lateral transfer of the fungus from *X. glabratus* [44,45,46,47]. Typically, each species of ambrosia beetle is associated with one or two primary fungal symbionts that are transmitted vertically to their progeny [46,48,49,50], and the progeny acquire these symbionts from their natal galleries while they feed and develop on these fungi. Within a host tree several different species of ambrosia beetles can breed sympatrically [51], providing an environment where beetles can acquire the fungal symbionts from other species. Although *X. glabratus* is still regarded as the most efficient vector, *H. lauricola* has been recovered from nine additional ambrosia beetle species in Florida, including *Ambrosiodmus lecontei* Hopkins, *Xyleborinus andrewesi* (Blandford), *Xyleborinus gracilis* (Eichhoff), *Xyleborinus saxesenii* (Ratzeburg), *Xyleborus affinis* Eichhoff, *Xyleborus bispinatus* Eichhoff, *Xyleborus ferrugineus* (Fabricius), *Xyleborus volvulus* (Fabricius), and *Xylosandrus crassiusculus* (Motschulsky) [44,45,46,47,52]. Of these species, it has been demonstrated under greenhouse conditions that *X. bispinatus* [52], *X. volvulus*, and *X. ferrugineus* [44] can transmit *H. lauricola* to healthy avocado trees and induce laurel wilt. Once established in a susceptible environment, laurel wilt is logistically difficult and labor intensive to manage [53], highlighting the importance of effective monitoring for early detection of the beetle vectors.

Most ambrosia beetles function ecologically as decomposers, colonizing stressed or dying trees [51], and ethanol-based lures are used in monitoring and surveillance programs for these species [54]. Ethanol is a natural byproduct of dead and fermenting wood or stressed tree tissues [54], and dispersing females utilize it as a kairomone for location of suitable hosts [55]. In contrast, *X. glabratus* can function as a primary colonizer, attacking healthy unstressed trees; consequently, it is not attracted to ethanol-based lures [9,56,57]. Instead, female *X. glabratus* are attracted to volatile monoterpenes and sesquiterpenes emitted from host wood and utilize these odor cues to find suitable trees for colonization [56,58,59,60].

Early field trapping studies relied on wounded redbay bolts to attract *X. glabratus* [9], and steam distillation of redbay wood and bark indicated high levels of two sesquiterpenes, α-copaene and calamenene. Subsequent field trials showed that manuka and phoebe essential oils, also high in these sesquiterpenes, were as attractive as redbay bolts to *X. glabratus* in South Carolina and Georgia [56]. However, in Florida field trials, manuka and phoebe oil lures captured many non-target Scolytinae, and manuka oil lures had a very short field life and were not competitive with *Persea* wood [58]. Additionally, commercial production of phoebe oil was discontinued due to depletion of the source trees in Brazil, thereby spurring research to identify an alternative attractant. In a comparison of seven essential oils, cubeb oil, naturally high in α-copaene, α-humulene, and β-caryophyllene, was identified as an improved attractant for *X. glabratus* [57]. In a concurrent study evaluating attraction and boring preferences of *X. glabratus* to wood from nine lauraceous species, emissions of α-cubebene, α-copaene, α-humulene, and calamenene were positively correlated with female attraction, and electroantennography confirmed antennal chemoreception of these host kairomones [61].

The most attractive synthetic lure currently used in monitoring and surveillance programs for *X. glabratus* consists of a distilled essential oil product enriched to contain 50% content of (-)-α-copaene [62,63]; its predecessor, the cubeb oil lure, contains a lower percentage of α-copaene, but is rich in several other sesquiterpenes. The current standard for monitoring overall ambrosia beetle populations consists of ethanol-based lures. However, little information exists about the efficacy of essential oil lures for detection of other ambrosia beetle species. Likewise, little is known about potential interactions between essential oils (sesquiterpenes) and ethanol. We hypothesize that many species may utilize both semiochemicals as part of their host location process: terpenoid emissions to assess the suitability of a host tree species, and ethanol emissions to determine the level of stress or morbidity of that host. In advance of the continued spread of *X. glabratus* into the western US and Mexico, where there are 120 reported species within the Lauraceae [64] and a $2.4 billion dollar avocado industry [65], the current work was initiated to better understand the attractiveness of essential oil lures, alone and in combination with ethanol, to ambrosia beetle communities in both agricultural (avocado grove) and forest (swampbay stand) ecosystems impacted by laurel wilt in Florida.

## 2. Materials and Methods

### 2.1. Lures

Sampling lures consisted of three commercial formulations: (1) low-release ethanol sleeve, 15 mL in a 40 cm long white plastic tube (Contech Enterprises Inc., Victoria, BC, Canada), (2) cubeb bubble lure, and (3) 50% α-copaene bubble lure, each containing 2 mL essential oil product in a 29-mm diam. clear plastic dispenser (products #3087 and #3302, respectively; Synergy Semiochemicals Corp., Delta, BC, Canada). In addition, one treatment used host wood as bait, consisting of bolts (42 cm length, 10 cm width) of silkbay (*Persea humilis* Nash) collected from Archbold Biological Station, Lake Placid, FL. At the time of bolt collection, the cut ends were wrapped in Parafilm M (Bemis Flexible Packaging, Neenah, WI, USA) to minimize desiccation and loss of volatile terpenoids prior to deployment in field tests.

### 2.2. Trap Design

Traps were constructed using two white sticky panels (23 × 28 cm, Scentry wing trap bottoms; Great Lakes IPM, Vestaburg, MI, USA) oriented back-to-back and suspended from an S-shaped wire hook. Ethanol sleeves were attached to the wire stem and then secured along the side of the panel to minimize contact with the adhesive surface (Figure 1A,B); bubble lures were clipped to the wire just above the sticky cards (Figure 1B). With silkbay, a wire loop was stapled to the top of the bolt, the bolt was hung vertically from the hook, and the sticky panels stapled to the bottom of the bolt (Figure 1C). The final assembly was topped with an inverted plastic plate to provide a protective covering. In previous field trials, this sticky trap design has been shown to be more effective for detection of *X. glabratus* [66] and other Scolytinae [41] than comparably baited funnel traps or cross-vane traps.

### 2.3. Field Tests

Parallel 7-week field tests were conducted at two Florida sites with laurel wilt. The first test was deployed in a commercial avocado grove in the Redland agricultural area of Miami-Dade County (25°35.530′ N, 80°27.509′ W), and ran from 3 February to 24 March 2015. The second test was conducted from 11 March to 29 April 2015 on a private ranch with ample woodlands containing swampbay trees in Highlands County (27°03.026′ N, 81°20.159′ W). Both tests evaluated captures of bark and ambrosia beetles with seven treatments: α-copaene, cubeb, ethanol, α-copaene + ethanol, cubeb + ethanol, silkbay wood, and a non-baited control trap. At test deployment, to promote release of host volatiles, a thin layer was cut off the ends of each silkbay bolt using a battery-operated reciprocating saw (Craftsman; Sears, Roebuck and Co., Chicago, IL, USA).

Both tests followed a randomized complete block design, with five replicate blocks. Each block was comprised of a row of traps hung 1–1.5 m above ground [67] in shaded locations. There was a minimum spacing of 10 m between adjacent traps within a row, and 30 m spacing between replicate rows. Traps were serviced weekly; at each sampling date, sticky panels were collected and replaced, a thin layer removed from each silkbay bolt, and traps rotated sequentially within each row. At the end of the 7-week tests, each treatment had rotated through each of the seven field positions within each row, thereby minimizing positional effects on beetle captures.

Samples were sorted under a stereo microscope in the laboratory (swampbay collections at the Subtropical Horticulture Research Station, Miami, FL; avocado collections at the University of Florida Tropical Research and Education Center, Homestead, FL, USA). All specimens of Scolytinae and Platypodinae were removed from the sticky panels, soaked in histological clearing agent (Histo-Clear II; National Diagnostics, Atlanta, GA, USA) to remove adhesive, and stored in 70% ethanol. Except for *Hypothenemus* spp. (the pygmy wood borers), all beetles were identified to species level according to previously described methods [23,68,69].

### 2.4. Statistical Analysis

One-way analysis of variance (ANOVA) was used to test the effect of treatment on mean field captures (beetles/trap/week). Significant ANOVAs were then followed by mean separation with Tukey HSD test. When necessary, data were either square-root (x + 0.05)-transformed or log (x + 1)-transformed to stabilize variance prior to analysis. Analyses were performed using SigmaPlot 14.0 (Systat Software Inc., San Jose, CA, USA). Results are presented as mean ± SEM; probability was considered significant at a critical level of α = 0.05.

## 3. Results

The combined sampling from both field tests resulted in the detection of 20 species of bark and ambrosia beetles, with the majority of captures representative of the scolytine tribe Xyleborini (Table 1). At the swampbay site, a total of 2314 specimens were collected. The most abundant beetles were *Xyleborinus andrewesi* which comprised 49.2% of the captures, followed by *Xyleborinus saxesenii* at 20.5% and *Xyleborus glabratus* at 11.0%. Equivalent trapping efforts in the avocado grove yielded 52,754 specimens. *Xyleborinus saxesenii* predominated in the avocado grove, accounting for 88.2% of the captures. Other species caught in high numbers (>400 specimens) included *Xylosandrus crassiusculus* (6.5%), *Hypothenemus* spp. (1.6%), *Xyleborus affinis* (1.2%), *Xyleborus volvulus* (Fabricius) (1.1%), and *Euwallacea perbrevis* (Schedl) (0.8%). The latter species and *Theoborus ricini* (Eggers) were only detected at the avocado site. In addition, *X. glabratus* and *X. andrewesi* were very minor components of the avocado beetle community, represented by only 5 and 2 captures, respectively, over the course of the 7-week test.

In the swampbay test (Figure 2), there were significant differences in mean captures of bark and ambrosia beetles (all species combined) among the treatments (F = 28.094; df = 6,28; *p* < 0.001). Traps baited with α-copaene + ethanol captured the highest number of beetles, and these were significantly higher than captures with ethanol alone. Captures with cubeb + ethanol were comparable to those of α-copaene + ethanol, but not greater than captures with ethanol alone. Mean captures with α-copaene alone, cubeb alone, and silkbay bolts were not significantly different than those of the non-baited control trap.

Analysis of captures of the three most abundant species at the swampbay site also indicated significant differences among treatments: *X. andrewesi* (F = 53.844; df = 6,28; *p* < 0.001), *X. saxesenii* (F = 32.324; df = 6,28; *p* < 0.001), and *X. glabratus* (F = 24.592; df = 6,28; *p* < 0.001). With *X. andrewesi* (Figure 3A), highest captures were observed in traps baited with α-copaene + ethanol and with cubeb + ethanol, and these captures were significantly greater than captures with ethanol alone or with any other treatment. Traps baited with ethanol alone caught more than the control, but this was not the case with α-copaene alone, cubeb alone, or silkbay bolts. With *X. saxesenii* (Figure 3B), equally high numbers were captured with all treatments that contained ethanol (i.e., ethanol alone, α-copaene + ethanol, and cubeb + ethanol); these numbers were significantly higher than those observed with any other treatment. In contrast, traps baited with ethanol alone captured the lowest number of *X. glabratus* (Figure 3C). Highest captures of this species were obtained with α-copaene alone, and the combination of α-copaene + ethanol resulted in a significant decrease in captures. Mean numbers caught with α-copaene + ethanol, cubeb alone, cubeb + ethanol, and silkbay bolts were all equivalent, and significantly greater than ethanol alone or the non-baited control trap.

In the avocado field test (Figure 4), there were significant differences in mean captures of bark and ambrosia beetles (all species combined) among the seven treatments (F = 30.394; df = 6,28; *p* < 0.001). Highest captures were obtained with traps that included ethanol (i.e., ethanol alone, α-copaene + ethanol, and cubeb + ethanol), and these numbers were significantly greater than those observed with any other treatment.

Analysis of captures by species also indicated significant differences among treatments at the avocado site: *X. saxesenii* (F = 30.558; df = 6,28; *p* < 0.001), *X. crassiusculus* (F = 30.380; df = 6,28; *p* < 0.001), *Hypothenemus* spp. (F = 11.075; df = 6,28; *p* < 0.001), *X. affinis* (F = 13.364; df = 6,28; *p* < 0.001), *X. volvulus* (F = 15.682; df = 6,28; *p* < 0.001), and *E. perbrevis* (F = 29.072; df = 6,28; *p* < 0.001). Consistent with the test in swampbay, captures of *X. saxesenii* (Figure 5A) were equally high with all treatments that contained ethanol (alone or in combination with α-copaene or cubeb oil), and were significantly higher than captures with all other treatments. With *X. crassiusculus* (Figure 5B), highest numbers were caught with cubeb + ethanol, which were significantly higher than ethanol alone, but not greater than captures with α-copaene + ethanol. Captures with α-copaene alone, cubeb alone, and silkbay bolts were no different than those of the non-baited control. With *Hypothenemus* spp. (Figure 5C), highest captures were obtained with all treatments that included ethanol (alone or in combination with essential oil lures), and captures were significantly greater than those with all other treatments. With *X. volvulus* (Figure 5D), traps baited with cubeb + ethanol captured the highest numbers, which were significantly greater than ethanol alone, α-copaene + ethanol, or other treatments. With *X. affinis* (Figure 5E), results were similar to those observed with *X. volvulus*; cubeb + ethanol captured the highest numbers, which were significantly greater than ethanol alone; however, not greater than those obtained with α-copaene + ethanol. In contrast to the other dominant species present at the avocado site, highest numbers of *E. perbrevis* (Figure 5F) were obtained with α-copaene alone, and these numbers were significantly higher than captures with any other treatment. Captures with ethanol alone were no different than those of the non-baited control, and addition of ethanol to α-copaene resulted in a significant reduction in captures.

## 4. Discussion

Host-derived volatile cues play an important role in the host selection and colonization process of ambrosia beetles [55,57,60,70,71,72,73]. Female beetles navigate through complex odor environments containing both host and non-host volatile compounds [74,75], and rely on a suite of compounds as ‘volatile signatures’, not single compounds alone, to identify suitable hosts from non-hosts [57,61,70,73,76,77]. For example, ambrosia beetles that colonize coniferous trees are generally attracted to volatile conifer terpenoids [51], but can detect, and are deterred by, some volatile angiosperm odors [78]. Understanding the host odors, or volatile signatures, used by ambrosia beetles to find suitable hosts can inform the development of attractive lures for monitoring species composition and population levels in specific environments. Further, understanding the temporal distribution and spatial patterns for a specific pest species are important when developing appropriate monitoring programs for a target pest [79].

In the current trapping study, beetle assemblages at the swampbay and avocado sites varied both in their species composition and population levels. Of particular note, captures of scolytine beetles were more than 20 times higher in the avocado grove, a large monoculture of trees exhibiting various stages of wilt-induced stress. *Xyleborinus saxesenii* predominated at the avocado site, comprising 88% of the captures, and *X*. *andrewesi* was the most abundant species in swampbay, making up 49% of the captures. *Xyleborinus andrewesi* was first detected in North America in 2010 in Lee County, FL [80] and is now established throughout the state. Traps baited with α-copaene + ethanol and cubeb + ethanol captured the most beetles overall in swampbay, and in avocado equally high numbers of total beetle captures were recovered from traps baited with ethanol, α-copaene + ethanol, and cubeb + ethanol. *Theoborus ricini* and *E. perbrevis* (previously *E*. nr. *fornicatus*) were captured exclusively at the avocado site. *Euwallacea perbrevis* is the tea shot-hole borer, vector of *Fusarium* dieback, another vascular disease of avocado now prevalent in south Florida [81]. Only 91 of the total 953 *Hypothenemus* spp. were captured at the avocado site. Although not identified to species level in this study, *Hypothenemus* members likely include eight species identified in a previous south Florida survey [82].

The highest numbers of *X. glabratus* were captured in traps baited with the α-copaene lure at the swampbay location, and very few were captured at the avocado site. This confirms previous trapping studies showing that the enriched α-copaene lure is the most attractive lure available for *X. glabratus* [62,63]. Like *X. glabratus*, *E. perbrevis* can colonize apparently healthy trees, and was most attracted to traps baited with the α-copaene lure. The addition of α-copaene to standard quercivorol (*p*-menth-2-en-1-ol, a fungal volatile) lures increases captures of *E. perbrevis* compared to quercivorol alone [42], but in the current study, traps baited with α-copaene alone captured significant numbers of *E. perbrevis*. Addition of ethanol to the copaene lure resulted in a significant reduction in captures of both *X. glabratus* and *E. perbrevis*, indicating a repellent effect on these major pest species. With *X. andrewesi*, low numbers were captured with ethanol alone or essential oils alone; however, both combinations of essential oil plus ethanol resulted in synergistic attraction and significantly higher captures, suggesting this species relies on both semiochemicals for host discrimination. *Xyleborinus saxesenii* captures were high with any treatment that included ethanol, but low in traps baited with essential oil lures alone, confirming previous results that ethanol is sufficient for detection and monitoring of *X. saxesenii* [83]. Similarly, the addition of (-)-α-pinene, a host-derived monoterpene emitted by some pine trees [84], to traps baited with ethanol does not increase captures of *X. saxesenii* compared to ethanol alone [85,86]; however, increasing release-rates of ethanol results in increased *X. saxesenii* captures [87].

Traps baited with ethanol and cubeb + ethanol captured the most *Hypothenemus* spp. Combining α-copaene with ethanol slightly reduced *Hypothenemus* spp. captures compared to ethanol alone, suggesting that α-copaene may function as a non-host odor for *Hypothenemus* spp. Cubeb + ethanol baited traps captured greater numbers of *X. crassiusculus, X. affinis*, and *X. volvulus* than either cubeb or ethanol alone, consistent with previous findings that cubeb alone is not attractive to these species [83]. Cubeb + ethanol baited traps also captured more *X. crassiusculus, X. affinis*, and *X. volvulus* compared to traps baited with α-copaene + ethanol, suggesting that volatile sesquiterpenes other than α-copaene (i.e., α-cubebene, α-humulene, and calamenene [83]) may interact with ethanol resulting in increased attraction. Cubeb + ethanol baited traps also captured large numbers of Scolytinae (species not identified) in previous trapping studies in Florida [57], and the current field results may provide information about the assemblage of species captured in that study. Ethanol baited traps are commonly used to capture *X. affinis* and *X. crassiusculus* [87,88,89,90], and the addition of (-)-α-pinene to ethanol baited traps increased trap captures of *X. affinis* in Florida compared to ethanol alone [85]; but the addition of (-)-α-pinene to ethanol baited traps decreased trap capture of *X. crassiusculus* compared to ethanol alone [85].

The most abundant species of Scolytinae captured in this study are not native to North America. *Xyleborus andrewesi* is native to the Old World tropics [91], *X. affinis* is native to South America [23,51], *X. volvulus* is native to South America and the Caribbean [51], and *X. saxesenii* and *X. crassiusculus* are native to Eurasia [51]. Invasive pests and pathogens can devastate native forest and agricultural systems leading to rapid environmental degradation [4,92]. Laurel wilt is currently altering the structure of affected native plant and agricultural communities in the southeastern USA and poses an imminent threat to other areas that have susceptible laurel species. High capture rates of *X. andrewesi, X. saxesenii*, and *X. glabratus* at the swampbay site were associated with the rapid spread of laurel wilt in natural forest ecosystems. Low capture rates of *X. glabratus* at the avocado site and high captures of *X. saxesenii, X. crassiusculus, X. affinis*, and *X. volvulus* were observed in laurel wilt affected avocado orchards in Florida. *Xyleborus bispinatus*, the species most frequently and persistently associated with *H. lauricola* in avocado systems [93], was found in relatively low numbers at both the avocado and swampbay sites. However, in another field study using ethanol-baited traps, *X. bispinatus* was one of the dominant ambrosia beetles in a Florida avocado grove [41]. Information about the diversity and population levels of these non-*X. glabratus* Xyleborini are thus important for laurel wilt management in both forest and agricultural environments.

Identifying the best lures, or lure combinations, for different ambrosia beetle species may aid the development of predictive risk models, and optimize current monitoring lures, for future invasions into naïve and susceptible landscapes. This trapping study provides further evidence to support the continued use of traps baited with enriched α-copaene lures for the detection and monitoring of the primary vector of laurel wilt, *X. glabratus*. Additionally, the beetle assemblages captured in this study provide evidence that the best lure combination for *X. andrewesi, X affinis, X. volvulus*, and *X. crassiusculus* is a combination of cubeb + ethanol; and that ethanol alone is sufficient for attracting and capturing *X. saxesenii*.

## 5. Conclusions

In parallel trapping tests directly comparing the communities of bark and ambrosia beetles resident in forest and agricultural ecosystems with laurel wilt, differences were observed in the species diversity and abundance. Total numbers captured in an avocado grove were more than 20 times greater than those intercepted in a swampbay forest. The dominant species detected in swampbay were *X. andrewesi*, *X. saxesenii*, and *X. glabratus*; major species in avocado included *X. saxesenii, X. crassiusculus, Hypothenemus* spp., *X. affinis*, and *X. volvulus*. Except for *Hypothenemus* (a group of bark beetles), all these beetles can vector *H. lauricola* and contribute to the spread of laurel wilt. Although Scolytinae are typically regarded as either primary colonizers (attracted to host terpenoids) or secondary colonizers (i.e., decomposers; attracted to ethanol), results of this study suggest that there is a continuum across this taxon. Standard ethanol lures were sufficient for detection of *X. saxesenii* and *Hypothenemus* spp., and α-copaene lures were sufficient for *X. glabratus* and *E. perbrevis*. However, species like *X. andrewesi*, *X. affinis*, *X. volvulus*, and *X. crassiusculus* were most attracted when ethanol and cubeb oil were presented in tandem. This behavior indicates that some ambrosia beetles utilize multiple kairomones for reliable host location. Improvement in pest detection programs may be achieved through novel combinations of attractants, and optimization of release rates, tailored for individual target species.

## Figures and Tables

**Figure 1 insects-13-00971-f001:**
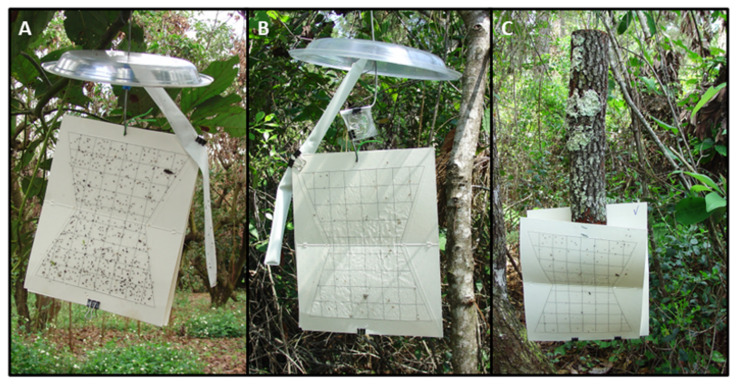
Sticky trap designs used to sample bark and ambrosia beetle communities at sites with laurel wilt. Representative photos depicting traps baited with (**A**) a low-release ethanol sleeve, (**B**) a combination of ethanol and essential oil lure, and (**C**) a bolt of host wood (silkbay, *Persea humilis*). Traps in panel (**A**) were deployed in an avocado grove in Miami-Dade County, FL; traps in panels (**B**,**C**) were deployed in a swampbay forest in Highlands County, FL.

**Figure 2 insects-13-00971-f002:**
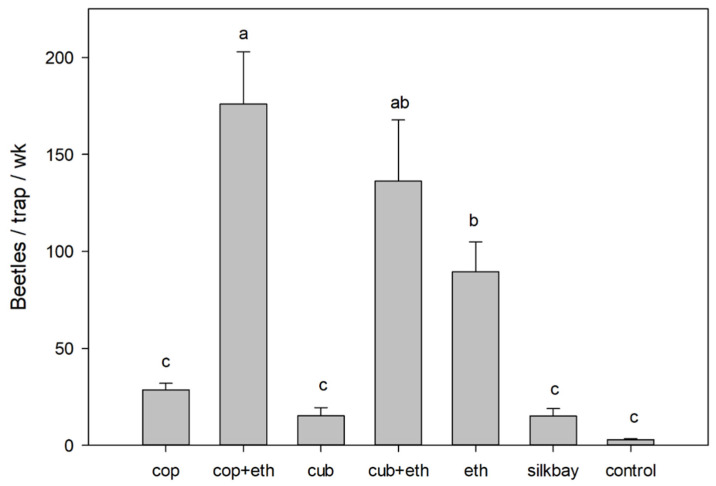
Mean (±SEM) captures of bark and ambrosia beetles (all species summed) in a 7-wk field test conducted in a swampbay forest with laurel wilt, Highlands County, Florida. Treatments contained α-copaene lures (cop), cubeb oil lures (cub), low-release ethanol lures (eth), combinations of essential oil lures and ethanol lures (cop + eth; cub + eth), bolts of silkbay (*Persea humilis*), and non-baited traps (control). Bars topped with the same letter are not significantly different (Tukey HSD mean separation, *p* < 0.05).

**Figure 3 insects-13-00971-f003:**
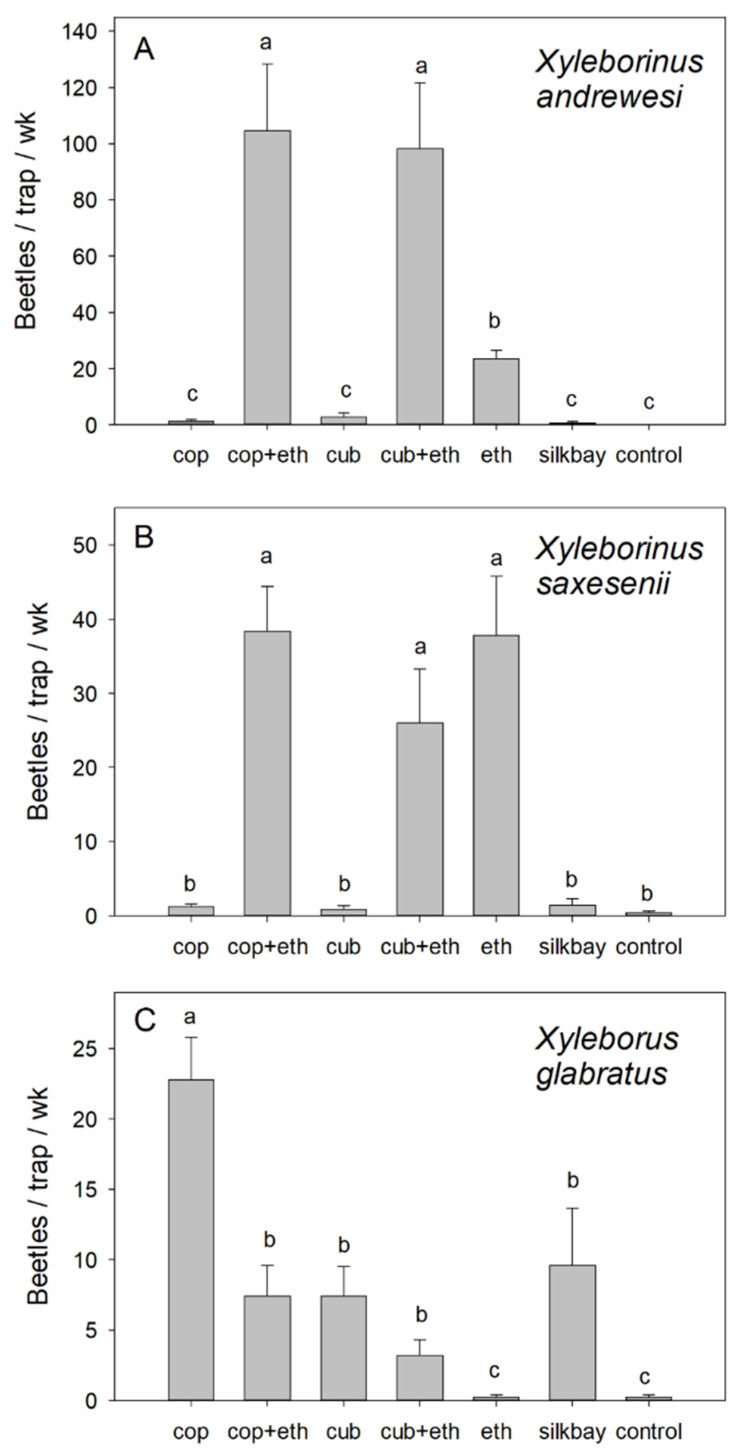
Mean (±SEM) captures of (**A**) *Xyleborinus andrewesi*, (**B**) *Xyleborinus saxesenii*, and (**C**) *Xyleborus glabratus*, the three dominant species sampled in a swampbay forest with laurel wilt, Highlands County, Florida. Treatments consisted of white sticky panel traps baited with 50% α-copaene lures (cop), cubeb oil lures (cub), low-release ethanol lures (eth), combinations of essential oil lures and ethanol lures (cop + eth; cub + eth), bolts of silkbay (*Persea humilis*), and non-baited traps (control). Bars topped with the same letter are not significantly different (Tukey HSD mean separation, *p* < 0.05). Please note differences in the scale of the *y*-axis.

**Figure 4 insects-13-00971-f004:**
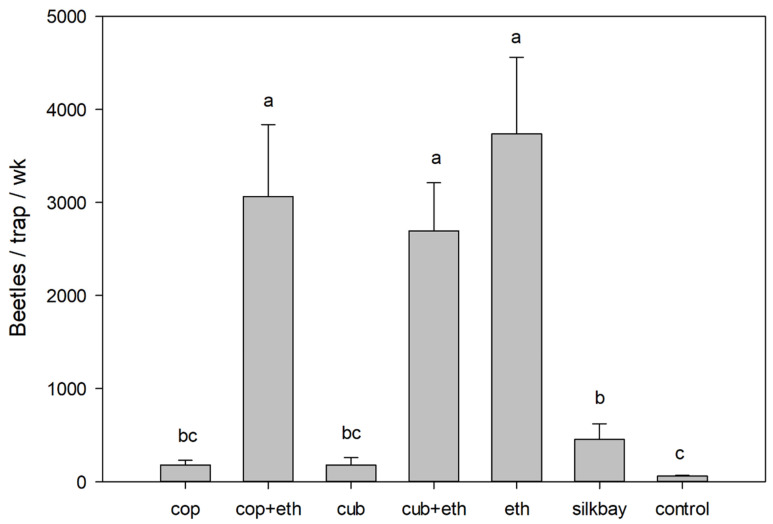
Mean (±SEM) captures of bark and ambrosia beetles (all species summed) in a 7-wk field test conducted in a commercial avocado grove with laurel wilt, Miami-Dade County, Florida. Treatments consisted of white sticky panel traps baited with 50% α-copaene lures (cop), cubeb oil lures (cub), low-release ethanol lures (eth), combinations of essential oil lures and ethanol lures (cop + eth; cub + eth), bolts of silkbay (*Persea humilis*), and non-baited traps (control). Bars topped with the same letter are not significantly different (Tukey HSD mean separation, *p* < 0.05).

**Figure 5 insects-13-00971-f005:**
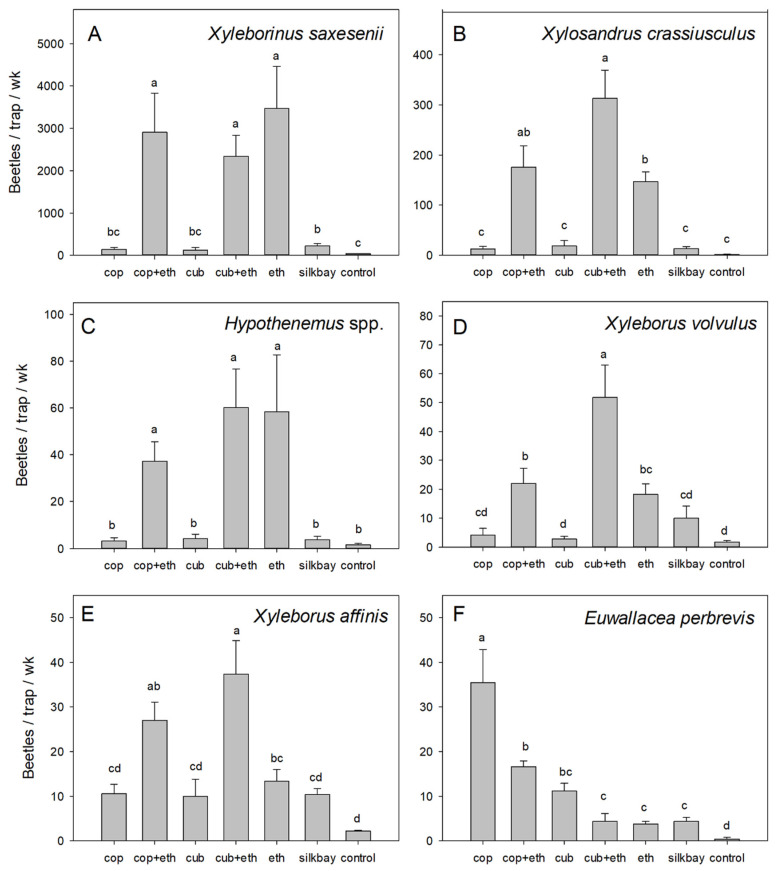
Mean (±SEM) captures of (**A**) *Xyleborinus saxesenii*, (**B**) *Xylosandrus crassiusculus*, (**C**) *Hypothenemus* spp., (**D**) *Xyleborus volvulus*, (**E**) *Xyleborus affinis*, and (**F**) *Euwallacea perbrevis*, the most abundant species sampled in an avocado grove with laurel wilt, Miami-Dade County, Florida. Treatments consisted of white sticky panel traps baited with 50% α-copaene lures (cop), cubeb oil lures (cub), low-release ethanol lures (eth), combinations of essential oil lures and ethanol lures (cop + eth; cub + eth), bolts of silkbay (*Persea humilis*), and non-baited traps (control). Bars topped with the same letter are not significantly different (Tukey HSD mean separation, *p* < 0.05). Please note differences in the scale of the *y*-axis.

**Table 1 insects-13-00971-t001:** Total captures of bark and ambrosia beetles (Coleoptera: Curculionidae) in parallel field tests at two Florida sites with laurel wilt: a swampbay forest in Highlands County and a commercial avocado grove in Miami-Dade County.

Species	Swampbay	Avocado
Subfamily Scolytinae		
Tribe Xyleborini		
*Ambrosiodmus devexulus* (Wood)	21	51
*Ambrosiodmus lecontei* Hopkins *	77	54
*Euwallacea perbrevis* (Schedl)	0	402
*Premnobius cavipennis* Eichhoff	1	40
*Theoborus ricini* (Eggers)	0	19
*Xyleborinus andrewesi* (Blandford) *	1139	2
*Xyleborinus gracilis* (Eichhoff) *	16	42
*Xyleborinus saxesenii* (Ratzeburg) *	474	46,537
*Xyleborus affinis* Eichhoff *	23	653
*Xyleborus bispinatus* Eichhoff *	86	64
*Xyleborus ferrugineus* (Fabricius) *	2	2
*Xyleborus glabratus* Eichhoff *	254	5
*Xyleborus volvulus* (Fabricius) *	10	566
*Xylosandrus compactus* (Eichhoff)	36	11
*Xylosandrus crassiusculus* (Motschulsky) *	21	3425
Tribe Cryphalini		
*Cryptocarenus heveae* (Hagedorn)	1	0
*Hypothenemus* spp.	91	863
Tribe Corthylini		
*Corthylus papulans* Eichhoff	44	4
*Monarthrum mali* (Fitch)	1	0
Subfamily Platypodinae		
*Euplatypus parallelus* (Fabricius)	17	14

* Species from which *Harringtonia lauricola*, causal agent of laurel wilt, has been isolated [47].

## Data Availability

The data are available from the authors upon request.

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
