# Peer review of "Community of Bark and Ambrosia Beetles (Coleoptera: Curculionidae: Scolytinae and Platypodinae) in Agricultural and Forest Ecosystems with Laurel Wilt"

_insects, 2022, doi:10.3390/insects13110971_

Round 1

Reviewer 1 Report

The manuscript, Community of Bark and Ambrosia Beetles (Coleoptera: Curculionidae: Scolytinae and Platypodinae) in Agricultural and Forest Ecosystems with Laurel Wilt, is well written and the experiment appropriately designed and analyzed. There are minor editorial comments/corrections found within the pdf. Once those changes are made the paper should be ready for publication. 

Author Response

Please review the attachment.

Reviewer 2 Report

The authors of this manuscript attempt to develop a monitoring method based on different the attractant lures for estimating the pest population density and species-specific differences in three different ambrosia beetle species.

 I think it could be an interesting paper, and represents a scientific contribution related to the scope of the Insects journal. Nevertheless, there are few questions and corrections before the manuscript should be accepted for publications. I therefore recommend its publication with minor corrections.

The issues that need to be addressed are listed below:

1.        The peak activity varied depending on species, year so a main question is: Is the first emergence of the studied species similar? The authors should include in more detail previous studies.

2.       The development of appropriate monitoring population density, temporal distribution and fluctuation for pest, requires an understanding of the spatial pattern of the target insect, which may vary with many aspects. The authors should include in more detail previous studies, for example: Wardhaugh C. W. 2014.

3.       In Table 1, what does the asterisk mean?

4.       Usually the essential oil chemical composition show high source variations. It would be very important to know the chemical composition of the cubeb bubble lure to explain part of the results: Does cubeb bubble formulations contain (-)-α-Copaene

  Discussion:  Please break it up into sub-sections if this is helpful or sort out this section somehow, because it is difficult to understand.
